# Study on the Status of Health Service Utilization among 3–5 Years Old Left-Behind Children in Poor Rural Areas of Hunan Province, China: A Cross-Sectional Survey

**DOI:** 10.3390/ijerph16010125

**Published:** 2019-01-04

**Authors:** Yufeng Ouyang, Jiaojiao Zou, Meimei Ji, Yefu Zhang, Tong Yuan, Lina Yang, Qian Lin

**Affiliations:** 1Department of Nutrition Science and Food Hygiene, Xiangya School of Public Health, Central South University, 110 Xiangya Road, Changsha 410078, China; ouyangyufeng0102@foxmall.com or oyyf0102@csu.edu.cn (Y.O.); zjj170605@foxmail.com or zjj227@csu.edu.cn (J.Z.); 2Department of Nutrition Science and Food Hygiene, Xiangya School of Public Health, Central South University, 110 Xiangya Road, Changsha 410078, China; jimeimei1024@foxmail.com (M.J.); yefuzhang@foxmail.com (Y.Z.); yuantong168@foxmali.com (T.Y.)

**Keywords:** Left-behind children, health service utilization, caregiver, rural area, China

## Abstract

The left-behind children (LBC) in China generally refer to children who remain in rural regions under the care of kin members while their parents migrate to urban areas. Due to some reasons, e.g., poverty, poor transportation conditions, lack of health resources, and preschool child care, it is hard for preschool-aged rural LBC to obtain essential health services. Random cluster sampling was used to recruit the caregivers and all the 3–5-year-old LBC in two rural counties in Hunan Province. A questionnaire was used to collect data on LBC demographics via face-to-face interviews with the caregivers. Health service needs were evaluated by the two-week prevalence rate, while health service utilization was measured by the two-week physician visit rate. Of the 559 respondents in the study, the two-week prevalence rate was 44.2% and the two-week physician visits rate was 48.6%. Nearly half of the sick children did not go to a hospital, 45.7% self-treated, and 5.3% did not take any treatment. The utilization rates of “health check,” “eye exam,” and “hearing screening for 3-year-olds” were extremely low (57.3%, 29.3%, 18.7%). The utilization rate of health services for preschool LBC in poor rural areas was extremely low, which can affect the normal growth and development of children.

## 1. Introduction

In China, the construction of medical and health service systems has been progressing steadily. However, the progress of basic health services for children still includes imbalances between rural and urban areas [1]. A national survey in 2013 showed the two-week prevalence rate among children under 5 years in rural areas of central China was 10.3%, and the two-week physician visit rate was 12.8%, lower than the rate in urban areas (14.6%) [2]. Poverty, poor transportation conditions, and a lack of health resources leads to significant barriers to health service access among local children, especially for left-behind children (LBC). 

The LBC in China generally refer to children who remain in rural regions under the care of kin members while their parents migrate to urban areas, usually for economic reasons [3]. In many cases, these children are taken care of by their grandparents. In 2013, China had 61.025 million LBC in rural areas, including 23.42 million children aged 0–5 years. The proportion of the 0–5-year-old children was 38.37% of all rural children, having increased by 47.73% compared to 2005 [4]. Without parental care and preschool child care institutions, it is hard for preschool-aged rural LBC to obtain essential health services. During early childhood, humans develop more rapidly than at any other point in their life. At this time, preventative healthcare is easier. However, limited health service resources may result in missing out on the best prevention and treatment period for these diseases among rural LBC. The preschool LBC in poor rural areas should receive more public health attention.

At present, in China, there are some studies on the utilization of health services for LBC in rural areas, but most of them focus on hospital services and childhood immunizations. A study indicated that the proportion of LBC receiving health checks in western rural areas was only 48.2% [5], far lower than the average level in China (83.1%) [6]. There are few reports on the utilization of public health services, such as vision and hearing screenings, in rural areas and urban areas. 

This baseline study selected Pingjiang County and Fenghuang County in Hunan Province to investigate the health service needs and the health service utilization of rural preschool LBC and explore relevant factors. These counties are both nationally designated as poor and having a high LBC population [7]. Pingjiang County is mainly hilly with the permanent population of Han majority [8]. Fenghuang County is located in the mountainous area with complex topography; the overall economic situation is worse than Pingjiang County. Fenghuang county’s ethnic minorities account for 78.9% of the resident population, mainly Tujia and Hmong [9]. This study aims to inform future health policy for LBC in poor rural areas.

## 2. Materials and Methods 

This study was conducted as part of a baseline survey of “The impact of conditional cash transfer on the nutritional status and physical development of 3–5-year-old LBC in poor rural areas of China” [7]. 

### 2.1. Research Setting

Hunan Province, located in south-central China, has a population where more than half of the rural children are LBC. Fenghuang County in Xiangxi Tujia and Miao Autonomous Prefecture and Pingjiang County in Yueyang City were chosen as the settings for this research project, which represent a range of geographies for this project [7].

### 2.2. Sampling

Village inclusion criteria: Villages were included if they had a minimum of 15 LBC (aged 3–5 years old) living in poor households, defined as annual income <2300 Renminbi (RMB, Chinese currency, 1 RMB = 0.145 USD), had no kindergarten or care centers for LBC, and did not receive any funding or benefits from other sources, such as charities or non-governmental organizations (NGOs). 

Participant inclusion criteria: All the 3–5-years-old LBC and their caregivers in the selected villages were eligible. Eligibility criteria for households in the intervention villages were as follows: (1) Households caring for at least one LBC (3–5 years old); (2) “poor households,” defined as average per-capita annual income lower than 2300 RMB; (3) households did not receive benefits from other charities, NGOs (Non-Governmental Organizations), or other similar programs.

In this study, villages were used as the basic units for grouping. The local health departments of Fenghuang County and Pingjiang County assisted us in the random selection of villages. The baseline measurements, including anthropometric measures and blood tests, were performed for all eligible LBC whose caregivers consented in the 40 selected villages before randomisation. The random allocation sequence was produced by an independent statistician who was not involved in the study. Additionally, the allocation considered the distance between villages to avoid contamination. If two villages were found to be too close to each other (<5 km), the random sequences would be generated and grouped again [7].

Eventually, 132 villages in Fenghuang County and 72 villages in Pingjiang County met the inclusion criteria of this study. Each County randomly selected 20 villages, and each village randomly selected 15 LBC who met the inclusion criteria.

### 2.3. Recruitment

As we described in a previous article [7], eligible caregivers were identified with the assistance of a local village doctor. Phone calls were used to notify the caregivers to participate in the questionnaire surveys in village clinics, 1–2 weeks in advance of the investigation. Travel expenses (60 RMB, about 9 USD) were reimbursed for each caregiver. Individuals were free to ask questions following the explanation or quit the investigation at any time. 

### 2.4. Ethical Approval 

This research was approved by the independent ethics committee of the Institute of Clinical Pharmacology, Central South University (registered number: ctxy-140003) and registered in the China Clinical Trial Register (registered number: ChiCTR-TRC-14005117). Informed consents were obtained by caregivers and all information was kept strictly confidential.

### 2.5. Data Collection

Face-to-face interviews were used to collect information. Investigators were the village doctors or health staff from township hospitals. Each investigator was trained by nutrition and public health researchers from Central South University. Quality control and guidance personnel were present during the interview process and a trained director conducted further validity checks in order to guarantee the accuracy of the final completed questionnaires.

The baseline investigation was carried out between January and March 2015. Data collection included: (1) General characteristics of LBC and their caregivers; (2) LBC’s health status and health service utilization. The two-week prevalence rate was used to reflect health service needs, while the two-week visiting rate and participation in basic public health services were used to evaluate health service utilization. Immunization coverage and rates of vision screening, hearing screening, and growth monitoring were used to evaluate the utilization of primary public health services. We conducted questionnaire interviews with caregivers and compared children’s immunization certificates to obtain a realistic appraisal of local LBC’s childhood immunization situation. The rates of vision screening and hearing screening were self-reported by caregivers. 

### 2.6. Statistical Analysis 

EpiData 3.0 software (The EpiData Association, Odense, Denmark) was used for data entry and the IBM SPSS 18.0 software package (IBM Corp, Armonk, NY, USA) was used for data analysis. The statistical methods used in this research include statistical descriptions and chi-squared tests. Descriptive data were reported in the form of a percentage and *p* ≤ 0.05 was considered to be statistically significant.

## 3. Results

### 3.1. General Characteristics of Study Population

The socio-demographic characteristics of the study population are presented in Table 1. Of the 559 respondents in this study, most of the participants were ethnic minorities (75.5%) in Fenghuang County, while the majority of the participants were Han ethnicity (98.3%) in Pingjiang County. A comfortable majority of these children (69.3%) had siblings. The proportion of premature and low birthweight children in the two counties was 8.0% and 4.0%, respectively. Most of the LBC were left behind by their parents (75.3%). 

### 3.2. Health Service Needs

As shown in Table 2, the two-week prevalence rate was 44.2% (246/556). Fever/headache/cough caused by respiratory tract infection accounted for 68.0%, while abdominal pain and diarrhea caused by gastrointestinal infection accounted for 13.3%. Among ill LBC, 13.0% of them experienced symptoms twice or more. The two-week physician visits rate was 48.6%. Most of the ill children were treated in village clinics (52.9%, Table 3). As shown in Figure 1, 45.7% of the sick children did not go to the doctor and instead were self-treated, and 5.3% of them did not take any treatment. Most caregivers reported “they think hospital treatment is useless” (34.1%) or “too far away from the hospital” (34.1%) as the main reasons for why LBC did not see a doctor (Figure 2). In our study, sick children who lived less than 15 min from the nearest medical institution were more likely to visit the hospital (46.6% vs. 34.1%, *p* < 0.05).

### 3.3. Health Services Utilization

The utilization rates for “health check in 2014,” “eye exam in 2014,” and “hearing screening for 3-year-olds” were extremely low (57.3%, 29.3%, and 18.7%, respectively) (Table 4). The utilization rate among LBC of children’s basic health service had a significant difference between the two counties. Table 5 shows the social demographic characteristics differences between the two counties, i.e., ethnicity, only child status, region, age of caregiver, and socioeconomic status on the utilization of health services for LBC had statistical difference significance (*p* < 0.001, Table 5). According to Figure 3, the main reason for why LBC did not use basic public health services was that “caregivers do not know about these public health services” (68.4%). The two main “other reasons” were: “Town hospitals don’t have the above health services and facilities” (22.2%) and “time conflicts with children going to school” (13.9%). In Chinese National Basic Public Health Services Norm 2013, a 3–6-year-old child can receive an annual free health check in local health institutions, included as a basic health care. As is shown in Table 6, awareness that “my child can receive annual free health checks in local health institutions” was low (45.5%), and the awareness rate of Fenghuang County was lower than that of Pingjiang County (30.7% vs. 59.4%, *p* < 0.001).

## 4. Discussion

The baseline survey showed that the two-week prevalence rate among the LBC was 44.2% and the two-week physician visit rate was 48.6%, which was higher than the average proportion of children under 5 in the rural areas of central China in 2013 (10.3%, 12.8%) [1]. It can be seen that the health state of LBC in rural areas is poor. We found that some of the LBC had meals near pig pens and dry toilets, and most of them did not wash their hand before meals or after using the toilet. More specialized child care facilities and preschool child care institutions are needed to provide a clean and safe environment for preschool. An Indonesian study found that unclean living conditions increased children’s incidence of diarrhea [10]. Two interventions conducted in rural Bangladesh found that better living household hygiene decreased the children’s incidence by 27%–30% [11,12,13]. 

We found that 52.9% of the LBC received medical treatment in the nearest township hospital. Among the sick children, the proportion of LBC who chose “self-treatment” was 49.0% and “no treatment measures” was 5.3%, which were above the national average (14.1% and 1.4%) [1]. “Caregivers thought the treatment in hospitals was ineffective” and “Inconvenient transportation” were the two main barriers to the health service utilization of LBC. Low-education levels may affect caregivers’ ability to cultivate children scientifically and to access child health services [14,15,16]. When seeking health care in rural areas, people have to face the challenge of accessibility and high transportation cost [17]. Some studies have pointed out that “distance from medical institution” is one of the main obstacles to get health services for most residents in resource-limited environments (especially in rural areas with complex geographical locations) [18,19,20,21,22,23]. We found that “time to the nearest medical institution” has a significant impact on the “the two-week physician visit rate”. Poor transportation conditions were the barriers for LBC receiving medical treatment and accessing health care services. A study in southern Tanzania found that infant and child mortality was higher among families living distances more than 5 km away compared to those living <5 km from the nearest health facility [24]. Some studies also found that the distance from medical institutions and education levels are key factors affecting health care utilization in rural areas [25,26,27]. 

In China, the immunization certificate is a voucher for childhood immunization. It is also a must-have health certificate for children before entering school. In our study, the self-reported vaccination rate of LBC was above 98%. Most caregivers had the awareness that “childhood immunization is provided by the government for free.” It reflects that China has made great progress in childhood immunization, the most basic public health service. There are laws in China that mandate vaccination certificates for children, which ensures their statutory role and value. However, we found there were still some problems. For example, many caregivers did not realize the importance of the vaccination schedule, which has led to delayed vaccination or partly unvaccinated. We observed that the date of some LBC’s immunization certificates was not accurate, the vaccination service period of the village clinic was not completely consistent with the national standard, and the vaccination services in the village clinic (including pre-examination, observation, notification) were not standardized, among others. The possible reasons and solutions are needed to explored in future studies.

At present, the emphasis on basic public health services for children is far from enough in China. We found that caregivers believed that the health service was important to children’s health, but they did not know that children could receive these public health services for free (68.4%). The current situation of public health services for children, such as children’s vision and hearing screening, is rarely reported in rural areas or urban areas. Since 2011, the Chinese national basic public health service, including health check and eye exam, were provided free for children aged 0–6 years, and free hearing screening was also included since 2013.

Caregivers’ health consciousness and their attention to different public health services will affect their utilization rate. The more they recognize the health service, the higher their utilization rate will be. We found the utilization of the rural LBC’s public health services was poor. About 57% of the LBC received a “Health check” in 2014, including height and weight measurements, since most of the caregivers put more emphasis on children’s anthropometric measurements. However, the rate of “Children’s eye exam” and “Children’s hearing screening” was low. In our field trips, most of the township hospitals lacked hearing screening equipment. In addition, compared to other public health services, “children’s hearing screening” was most subject to time constraints and geographic conditions. Children can receive health checks or eye exams every year, but they can only receive hearing screening in a few fixed time periods. Hearing screenings cannot be performed effectively in time if the child is sick, the caregiver forgot, or it was inconvenient to visit township hospitals.

A primary care providing a vision or hearing screening is important for each child to receive further evaluation and necessary treatment. Amblyopia has become the most common visual disability in Chinese children. The prevalence rate of amblyopia in Chinese children was reported at 2.8% in 2016, with 6.16 million children aged 4–6 years [28]. Around 466 million people worldwide have a disabling hearing loss and 34 million of these are children [29]. At present, there are about 30,000 deaf children under the age of 7 in Hunan province, and they are increasing at an annual rate of about 1000. Among them, more than 8000 deaf children live in extremely poor families and generally lack basic rehabilitation and self-rescue ability [30]. The main causes of hearing loss in children are infections, such as mumps, meningitis, and chronic ear infections. Sixty percent of children’s hearing loss can be prevented by vaccinating against infectious diseases, early hearing screening, and treating children with chronic ear infections [31]. Three to five years of age is a sensitive period for children’s development, and it is also the best intervention time for vision and hearing impairment. Early detection and timely treatment are urgent for rural children. 

The health status and the public health services of LBC in Fenghuang County were significantly worse than those in Pingjiang County. Fenghuang County is located in a mountainous area, with complex geographical features and inconvenient transportation. The health care knowledge of the caregivers in Fenghuang County was lower than that of Pingjiang County. Low education levels and language barriers were prominent in these caregivers: Some elderly Hmong caregivers could neither speak nor understand Mandarin, and only a few local medical staff could speak Miao language. Most of the Hmong LBCs were living in scattered areas that were difficult to centrally manage. 

In summary, it is important to improve society’s emphasis on the accessibility of health services for LBC in remote rural areas. The health system itself is a social determinant of health [30]. Health systems can reduce inequalities in access to health services through a variety of strategic policies that can benefit the most vulnerable groups due to personal, social, and geographic factors. A study in northern Brazil showed that adopting a primary health care strategy basically achieved health care equality in the Sobral region through the family health plan [32]. Therefore, the health sector can adopt some policy measures, such as incorporating vision and hearing screening into preschool health examinations; encouraging the establishment of hospital groups, medical complexes, and counterpart support so that community medical institutions can improve the service capacity for regular visits; and providing medical facilities and equipment for rural areas. On the other hand, the development of new medical models, such as telemedicine and mobile van services in the form of “charity mobile clinics”, may improve the accessibility of children’s health care services. In rural India, local community-based mobile health interventions (ImTeCHO, Gujarat, India) are adopted to expand the coverage of local health services [33]. Bangladesh has integrated the MCH (maternal and child health) handbook with mobile text messages and audio calls as a mobile health care tool for MCH services [34]. Accredited Social Health Activists (ASHAs), a new village-based front line of health workers, has been created in India for primary health care [33]. Improving the attractiveness of rural grassroots pediatric medical staff and increasing the number of qualified medical workers have proven to be effective solutions. Granting funds to poor and remote areas where LBC are prevalent and that lack child care institutions, building kindergartens, and centrally managing preschool LBC are also effective means.

This study reports for the first time the utilization of health services for preschool LBC in poor rural areas. The results of the study can provide a reference for further adjustment of health service strategies and policies for LBC in poor rural areas. However, there were some limitations in this study. Some subjective sensory indicators were used in the questionnaire, which may lead to a reporting bias and affect the results of the survey. Some older Hmong caregivers cannot understand Mandarin and needed translation assistance. During this process, the meaning of some of the items in the questionnaire may have been misinterpreted. Third, limited by terrain and human resources, small sample sizes may result in insignificant associations between variables, requiring further larger sample sizes. Fourth, since this survey was launched in 2014, we are referring to the 2013 edition of the National Basic Health Service Code. LBC exist based on China’s unique societal context, and children’s basic health services in different countries contain different items. Although we have discussed some international research, most of the results are based on Chinese data.

## 5. Conclusions

The utilization rate of health services for preschool LBC in poor rural areas was extremely low, which can affect the normal growth and development of children. We urgently need to take implement innovative strategies and public health educational programs to improve the utilization of health services for preschool LBC.

## Figures and Tables

**Figure 1 ijerph-16-00125-f001:**
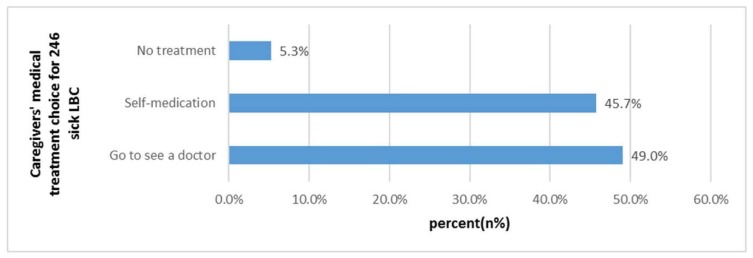
Caregivers’ medical treatment choice for 246 sick LBC (*n*, %).

**Figure 2 ijerph-16-00125-f002:**
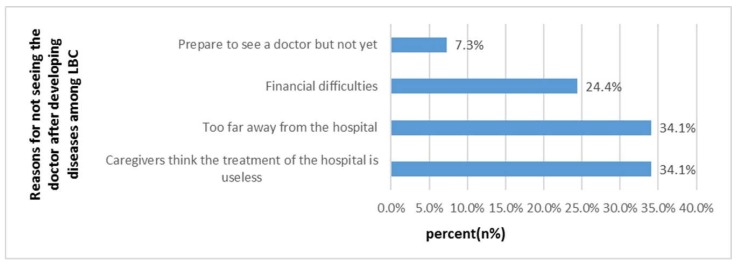
Reasons for why LBC did not see a doctor after developing diseases in the past two weeks (*n*, %).

**Figure 3 ijerph-16-00125-f003:**
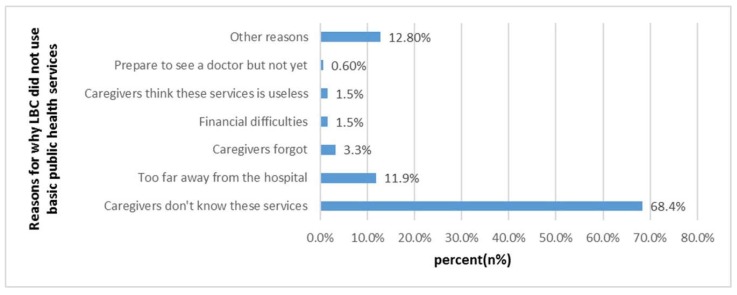
Reasons for why LBC did not use basic public health services (*n*, %).

**Table 1 ijerph-16-00125-t001:** Socio-demographic characteristics of left-behind children (*n*, %) (*n* = 559).

Characteristics	Fenghuang County (*n* = 273)	Pingjiang County (*n* = 286)	Total
			(*n* = 559)
Age (year)			
3~	87 (31.9%)	88 (30.8%)	175 (31.3%)
4~	89 (32.6%)	90 (31.4%)	179 (32.0%)
5~	97 (35.5%)	108 (37.8%)	205 (36.7%)
Sex			
Male	153 (56.0%)	144 (50.3%)	297 (53.1%)
Female	120 (44.0%)	142 (49.7%)	262 (46.9%)
Ethnicity			
Han	67 (24.5%)	281 (98.3%)	348 (62.3%)
Minorities	206 (75.5%)	5 (1.7%)	211 (37.7%)
The only child ^#^	57 (24.6%)	93 (36.3%)	150 (30.7%)
Premature ^#^	15 (6.9%)	22 (8.9%)	37 (8.0%)
Low birth weight (<2500 g) ^#^	7 (2.7%)	9 (4.2%)	16 (4.0%)
Breast feeding ^#^			
No	45 (20.2%)	77 (31.2%)	122 (26.0%)
Yes (<6 months)	9 (4.0%)	23 (9.3%)	32 (6.8%)
Yes (≥6 months)	169 (75.8)	147 (59.5%)	316 (67.2%)
Left-behind situation			
Father away	69(25.3%)	42 (14.7%)	111 (19.9%)
Mother away	14 (5.1%)	13 (4.5%)	27 (4.8%)
Both parents away	190 (69.6%)	231 (80.8%)	421 (75.3%)

^#^ Missing data for baseline investigation.

**Table 2 ijerph-16-00125-t002:** Health situation and health service needs among left-behind children (*n*, %) (*n* = 246).

Variables	Fenghuang County (*n* = 112)	Pingjiang County (*n* = 134)	Total (*n* = 246)
Two-week prevalence ^#^	112 (41.2%)	134 (47.2%)	246 (44.2%)
Two-week physician visiting ^#,^**	39 (34.8%)	80 (60.2%)	119 (48.6%)
Presenting symptoms(man-time) ^#^			
Fever/Headache/Cough	93 (76.2%)	96 (61.5%)	189 (68.0%)
Abdominal pain/Diarrhea	13 (10.7%)	24 (15.4%)	37 (13.3%)
Chest pain/Flustered/Palpitations	1 (0.8%)	2 (1.3%)	3 (1.1%)
Trauma	2 (1.6%)	1 (0.6%)	3 (1.1%)
Others	13 (10.7%)	33 (21.2%)	48 (16.5%)
Types of symptoms ^#^			
1	102 (91.9%)	116 (86.6%)	218 (89.0%)
2	8 (7.2%)	14 (10.4%)	22 (9.0%)
3 and above	1 (0.9%)	4 (3.0%)	5 (2.0%)

Chi-square test, ** *p* < 0.01; ^#^ Missing data for baseline investigation.

**Table 3 ijerph-16-00125-t003:** Visits to the hospitals at different levels (*n*, %) (*n* = 119).

Hospitals at Different Levels	Fenghuang County (*n* = 39)	Pingjiang County (*n* = 80)	Total (*n* = 119)	χ^2^	*p*
Village clinics	15 (38.5%)	48 (60.0%)	63 (52.9%)	7.611	0.107
Township hospitals	15 (38.5%)	19 (23.8%)	34 (28.6%)		
County hospitals	6 (15.4%)	7 (8.8%)	13 (10.9%)		
City hospitals and above	3 (7.7%)	3 (3.8%)	6 (5.0%)		
others	0 (0.0%)	3 (3.8%)	3 (2.5%)		

**Table 4 ijerph-16-00125-t004:** Basic public health services among left-behind children (*n*, %) (*n* = 557).

Answer	Fenghuang County (*n* = 273)	Pingjiang County (*n* = 284)	Total (*n* = 557)	χ^2^	*p*
Received the childhood immunization ^#^				5.328	0.070
yes	263 (98.1%)	283 (100.0%)	546 (99.1%)		
no	1 (0.4%)	0 (0.0%)	1 (0.2%)		
unclear	4 (1.5%)	0 (0.0%)	4 (0.7%)		
Received the child health handbook				226.046	<0.001
yes	40 (4.7%)	221 (77.8%)	261 (46.9%)		
no	144 (52.7%)	48 (16.9%)	192 (34.5%)		
unclear	89 (32.6%)	15 (5.3%)	104 (16.7%)		
Received free health check in 2014				80.584	<0.001
yes	104 (38.1%)	215 (75.7%)	319 (57.3%)		
no	141 (51.6%)	58 (20.4%)	200 (35.9%)		
unclear	28 (10.3%)	20 (7.0%)	38 (6.8%)		
Measured height and weight for free				85.696	<0.001
yes	92 (33.7%)	206 (72.5%)	298 (53.5%)		
no	147 (53.8%)	58 (20.4%)	205 (36.8%)		
unclear	34 (12.5%)	20 (7.0%)	54 (9.7%)		
Received free eye exam in 2014 ^#^				95.519	<0.001
yes	28 (6.6%)	135 (47.5%)	163 (29.3%)		
no	213 (78.3%)	121 (42.6%)	334 (60.1%)		
unclear	31 (11.4%)	28 (9.9%)	59 (10.6%)		
Received hearing screening at 3 years old ^#^				57.001	<0.001
yes	18 (6.6%)	86 (30.4%)	104 (18.7%)		
no	219 (80.5%)	153 (54.1%)	372 (67.0%)		
unclear	35 (12.9%)	44 (15.5%)	79 (14.2%)		

^#^ Missing data for baseline investigation.

**Table 5 ijerph-16-00125-t005:** Public health services for LBC with different social demographic characteristics (*n*, %).

Variables	Health Check in 2014	Eye Exam in 2014	Hearing Screening for 3-Year-Olds
Total (*n* = 559)	319 (57.0%)	163 (29.2%)	104 (18.6%)
LBC’s characteristics			
Age (year)			
3~	106 (60.6%)	59 (33.7%)	39 (22.3%)
4~	99 (55.9%)	44 (24.9%)	30 (17.0%)
5~	114 (55.6%)	60 (29.4%)	35 (17.2%)
Sex			
Female	164 (55.2%)	78 (26.3%)	48 (16.2%)
Male	155 (59.6%)	85 (32.8%)	56 (21.6%)
Ethnicity			
Han	202 (58.2%)	107 (30.8%)	80 (23.1%)
Minorities	117 (55.7%)	56 (26.8%)	24 (11.5%)
Only child^#^			
yes	83 (55.7%)	43 (28.9%)	37 (24.8%) ^**^
no	196 (58.2%)	100 (29.8%)	53 (15.8%)
Left-behind situation			
father work outside	58 (52.7%)	31 (28.2%)	15 (13.5%)
mother work outside	13 (48.1%)	7 (25.9%)	4 (14.8%)
parents work outside	248 (59.0%)	125 (29.8%)	85 (20.3%)
Caregiver’s characteristics			
Age (year) *			
20~	30 (50.0%)	16 (26.7%)	5 (8.3%)
40~	155 (59.2%)	73 (27.9%)	46 (17.6%)
60~	134 (57.0%)	74 (31.6%)	53 (22.6%)
Sex			
female	202 (55.6%)	104 (28.7%)	62 (17.6%)
male	117 (60.3%)	59 (30.4%)	42 (21.8%)
Occupation			
nonagricultural Worker	37 (58.7%)	19 (30.2%)	12 (19.0%)
farmer	282 (57.1%)	144 (29.2%)	92 (18.4%)
Education level			
no formal education	95 (54.9%)	50 (29.1%)	28 (16.3%)
primary school	153 (56.9%)	77 (28.5%)	53 (19.7%)
junior middle school	58 (62.4%)	31 (33.3%)	19 (20.4%)
high school and others	13 (61.9%)	5 (23.8%)	4 (19.0%)
Relation to LBC			
father/mother	38 (51.9%)	24 (32.4%)	8 (10.8%) *
grandfather	110 (61.5%)	59 (33.0%)	44 (24.7%)
grandmother	164 (56.0%)	79 (27.1%)	51 (17.5%)
others	7 (63.6%)	1 (9.1%)	1 (9.1%)
Areas			
mountainous area	156 (57.4%)	78 (28.8%)	33 (12.2%) **
hilly area	163 (57.2%)	85 (29.8%)	71 (24.9%)
Socioeconomic status ^&^			
low	83 (60.8%)	39 (28.3%)	15 (10.9%) **
middle	162 (57.9%)	81 (29.0%)	54 (19.4%)
high	74 (53.2%)	43 (30.9%)	35 (25.2%)

Chi-square test, * *p* < 0.05, ** *p* < 0.01; ^&^ Socioeconomic status, which was estimated following principal component analysis, including various items related to the economic status (annual per capita income, housing type, access to tap water, number of children at home, and number of bedridden patients at home). ^#^ Missing data for baseline investigation.

**Table 6 ijerph-16-00125-t006:** Awareness of two public health services among left-behind children’s caregivers (*n*, %) (*n* = 556).

Awareness	Fenghuang County (*n* = 270)	Pingjiang County (*n* = 286)	Total (*n* = 556)
Know children’s planned immunization is free **	199 (74.0%)	251 (88.4%)	450 (81.4%)
Know their child can receive free health check in local health institutions **	83 (30.7%)	170 (59.4%)	253 (45.5%)

Chi-square test, ** *p* < 0.01.

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
