# Peer review of "Study on the Status of Health Service Utilization among 3–5 Years Old Left-Behind Children in Poor Rural Areas of Hunan Province, China: A Cross-Sectional Survey"

_ijerph, 2019, doi:10.3390/ijerph16010125_

Reviewer 1 Report

Your article was well written and well organized. The article will fill a gap in the literature. In the future, you may consider a survey with just Hmong who may have a different worldview of health and healing.  

Author Response

Response to Reviewer 1 Comments

Point 1: Your article was well written and well organized. The article will fill a gap in the literature. In the future, you may consider a survey with just Hmong who may have a different worldview of health and healing. 

Response: We would like to thank you for those insightful comments and suggestions. The comments and suggestions have been extremely valuable and helpful for our future research and future health policy for LBC in poor rural areas. We may consider a survey with just Hmong in future studies. Here we submit a new version of our manuscript with the title “Study on the Status of Health Service Utilization among 3–5 years old Left-Behind Children in Poor Rural Areas of Hunan Province, China: A cross-sectional Survey,” (Manuscript ID: ijerph-408597), which has been revised according to the suggestions. Efforts were made to correct the mistakes and improve the English of the manuscript. We also have checked our database, repeated all the analysis, and revised mistakes. You will clearly see the difference made to the revised manuscript.

Reviewer 2 Report

This is a study assessing health care utilization among children raised by caregivers in two rural Counties in China. I have provided some comments/suggestions for the authors  to consider.

Abstract

·        Briefly define LBC.

Introduction

·        Please note in the first paragraph of the “materials and methods” why you chose to compare these two Counties. The authors describe the differences in the discussion, but don’t provide  any rationale at the beginning so that the readers know why these areas are being compared.

Methods

·         Please provide a reference for lines 64-65 regarding the population of LBC.

·        Please spell out RMB the first time it’s noted.

·        In lines 78-80, please describe the random selection process that each County underwent.

Results/Tables/Figures

·        Please double check the frequencies and percentages. For example, for gender in Fenghuang county, male and female adds up to 273, but the total at the top of the column says 279. If there is some missing data, please note that. Also, the percentages for gender in that county add up to 102%.

·        Beginning in line 117 the authors note the characteristics of  the caregivers, but there is no corresponding table with these data.

·        Table 4, for your p-values, please note a value of<0.0001 rather than 0.000.  

·        Line 148-150, this should not be described as significant impacts, but rather simply as differences.

·        For Table 5, when statistically significant differences are noted, are these statistically significant  for all services?

·        Figures, please change the x-axis label to say percent, rather than man-time.

Discussion

·        Please do not re-report the numbers from the results section.

·        The discussion section needs to be more focused and some of the paragraphs shortened.

Author Response

Response to Reviewer 2 Comments 

Dear reviewer,

We would like to thank you for those insightful comments and suggestions. The comments and suggestions have been extremely valuable and helpful for revising and improving our paper. Here we submit a new version of our manuscript with the title “Study on the Status of Health Service Utilization among 3–5 years old Left-Behind Children in Poor Rural Areas of Hunan Province, China: A cross-sectional Survey,” (Manuscript ID: ijerph-408597), which has been revised according to the suggestions. Efforts were made to correct the mistakes and improve the English of the manuscript. We also have checked our database, repeated all the analysis, and revised mistakes. You will clearly see the difference made to the revised manuscript.

Below we detail the point-to-point response to the reviewer’s comments.

Abstract:

Comment 1: Briefly define LBC

Response: Thanks for the reviewer's suggestion. We have defined the LBC in the Abstract ( Line 16-17, page 1).

Correction: The left-behind children (LBC) in China generally refer to children who remain in rural regions under the care of kin members while their parents migrate to urban areas.

Introduction:

Comment 2: Please note in the first paragraph of the “materials and methods” why you chose to compare these two Counties. The authors describe the differences in the discussion, but don’t provide any rationale at the beginning so that the readers know why these areas are being compared.

Response: Thanks for pointing out this. We have added the characteristics of Ethnicity, Landform and Socioeconomic status of the two Counties in Line 58-62, page 2. We chose these two representative counties to reflect the different kinds of barriers in accessing health services in rural China.

Correction: Pingjiang County is mainly hilly with the permanent population of Han majority [8]. Fenghuang County is located in the mountainous area with complex topography, the overall economic situation is worse than Pingjiang County. Fenghuang county's ethnic minorities account for 78.9 % of the resident population, mainly Tujia and Hmong [9]. This study aims to inform future health policy for LBC in poor rural areas.

Methods:

Comment 3: Please provide a reference for lines 64-65 regarding the population of LBC.

Response: Thanks for the reviewer's suggestion. We have added the No.7 reference about the population of LBC in Line 71, page 2.

Comment 4: Please spell out RMB the first time it’s noted.

Response: Thanks for pointing out this. Renminbi (RMB) is the official currency of China. We have spelt out RMB in Line 74-75, page 2.

Correction: ……<2300 Renminbi (RMB, Chinese currency, 1 RMB=0.145 USD),……

Comment 5: In lines 78-80, please describe the random selection process that each County underwent.

Response: Thank you for the comments. We have added the random selection process in Line 83-90, page 2.

Correction: In this study, villages were used as the basic units for grouping. The local health departments of Fenghuang County and Pingjiang County will assist us in the random selection of villages. The baseline measurements, including anthropometric measures and blood tests, will be performed for all eligible LBC whose caregivers have consented in the 40 selected villages before randomisation. The random allocation sequence will be produced by an independent statistician who is not involved in the study. Additionally, allocation will consider the distance between villages to avoid contamination. If two villages are found to be too close to each other (<5 km), the random sequences will be generated and grouped again [7].

Results/Tables/Figures:

Comment 6: Please double check the frequencies and percentages. For example, for gender in Fenghuang county, male and female adds up to 273, but the total at the top of the column says 279. If there is some missing data, please note that. Also, the percentages for gender in that county add up to 102%.

Response: Thanks for pointing out this. We have checked our database, repeated all the analysis, and revised mistakes. The Sample size in Fenghuang county and Pingjiang county were 273 and 286, respectively. The percentages for Male and Female were 56.0% and 44.0%, respectively. The revised details can be found in Table 1, page 3-4.

Comment 7: Beginning in line 117 the authors note the characteristics of the caregivers, but there is no corresponding table with these data.

Response: Thank you for the comments. We have deleted these sentences in Results. We noted the characteristics of the caregivers and quoted a reference (No.14 reference) in Line 196, page 8. This reference was about the LBC caregiver’s health service utilization in the same study.

Comment 8: Table 4, for your p-values, please note a value of<0.0001 rather than 0.000.

Response: Thanks for pointing out this. We have revised it. The revised details can be found in Table 4, page 3-4.

Comment 9: Line 148-150, this should not be described as significant impacts, but rather simply as differences.

Response: Thanks for pointing out this. We have modified the expressions of it in Line 161-163, page 6.   

Correction: Table 5 shows the social demographic characteristics differences between the two counties, i.e., Ethnicity, only child status, region, age of caregiver, and socioeconomic status on the utilization of health services for LBC had statistical difference significance (p < 0.001, Table 5).

Comment 10: For Table 5, when statistically significant differences are noted, are these statistically significant for all services?

Response: Thank you for the question. We have corrected it. These statistically significant only for Hearing screening for 3-year-olds. The revised details can be found in Table 5, page 6-7.

Comment 11: Figures, please change the x-axis label to say percent, rather than man-time.

Response: Thanks for the reviewer's suggestion. We have changed all the x-axis label in Figures.

Discussion

Comment 12: Please do not re-report the numbers from the results section.

Response: Thank you for the suggestions. Major revision was made in the discussion section. We have removed the numbers from the results section and have reedited the statements. The revised details can be found in the discussion section of our revision manuscript(page7-10).

Comment 13: The discussion section needs to be more focused and some of the paragraphs shortened.

Response: Thank you for the comments. We have made a major revision to the discussion section. We have reedited the statements in the discussion section to make them more concise and the topic more prominent. Paragraphs were reorganized in a more logical sequence. We moved lines 234-242(page 8, the original manuscript) about the current situation of basic public health services for children in China to lines 219-225(page 8, the revision manuscript). We removed lines 201 -202(page 8, the original manuscript) of the discussion section about LBC’s immunization coverage and added them to the section 2.5. Data Collection (lines 116-119, page 3, the revision manuscript). The revised details can be found in the discussion section of revision manuscript(page7-10).

Reviewer 3 Report

Authors suggeted that the utilization rate of health services for preschool LBC in poor rural areas was extremely low, which can affect the normal growth and development of children. This is very important to know not only clinician but researchers, however, with regard to health information of these research, I recommend to add or consider these previsoous studies. In addition, English language and style are fine/minor spell check required in the present study.

1)

Impact of Parents' Comprehensive Health Literacy on BMI in Children: A Multicenter Cross-Sectional Study in Japan.

Nakamura D, et al.

J Sch Health. 2018 Dec;88(12):910-916.

2)

Associations between Parents' Health Literacy and Sleeping Hours in Children: A Cross-Sectional Study.

Ogi H, et al. Healthcare (Basel). 2018 Apr 2;6(2).

Author Response

Comment: Authors suggested that the utilization rate of health services for preschool LBC in poor rural areas was extremely low, which can affect the normal growth and development of children. This is very important to know not only clinician but researchers, however, with regard to health information of these research, I recommend to add or consider these previous studies. In addition, English language and style are fine/minor spell check required in the present study.

Response: Thanks for the reviewer's helpful suggestion. We have added these two papers as No.15 and No.16 reference in Line 196, page 8. Efforts were made to correct the mistakes and improve the English of the manuscript.

Round  2

Reviewer 2 Report

The authors have adequately addressed most of my comments and requested revision. There is one small change I would recommend for the authors.

In describing the randomization approach in section 2.2, the authors use both future and past tense. Please use past tense for this description.